



# Relating snowfall observations to Greenland ice sheet mass changes: an atmospheric circulation perspective

Michael R. Gallagher[1,2], Matthew D. Shupe[1,2], Hélène Chepfer[3,4], and Tristan L'Ecuyer[5]

[1]Cooperative Institute for Research in Environmental Science, Boulder, Colorado, USA
[2]NOAA/Physical Sciences Laboratory, Boulder, Colorado, USA
[3]LMD/IPSL, Sorbonne Université, Paris, France
[4]LMD/IPSL, CNRS, Ecole Polytechnique, Palaiseau, France
[5]Department of Atmospheric and Oceanic Sciences, University of Wisconsin-Madison, Madison, WI, USA

**Correspondence:** Michael Gallagher (michael.r.gallagher@noaa.gov)

**Abstract.**

Snowfall is the major source of mass for the Greenland ice sheet but the spatial and temporal variability of its contributions to mass balance have so far been inadequately quantified. By characterizing local atmospheric circulation and utilizing CloudSat spaceborne radar observations of snowfall, we provide a detailed spatial analysis of snowfall variability and its relationship

to Greenland mass balance, presenting first-of-their-kind daily maps of the spatial variability in snowfall from observations across Greenland. For identified regional atmospheric circulation patterns, we show that the spatial distribution and net mass input of snowfall varies significantly with the position and strength of surface cyclones. Cyclones west of Greenland driving southerly flow contribute significantly more snowfall than any other circulation regime, with each daily occurrence of the most extreme southerly circulation pattern is contributes an average of 1.66 Gt of snow to the Greenland ice sheet. While cyclones

east of Greenland, patterns with the least snowfall, contribute as little as 0.58 Gt each day. Above 2 km on the ice sheet where snowfall is inconsistent, extreme southerly patterns are the most significant mass contributors, with up to 1.20 Gt of snowfall above this elevation. This analysis demonstrates that snowfall over the interior of Greenland varies by up to a factor of five depending on regional circulation conditions. Using independent observations of mass changes made by the Gravity Recovery and Climate Experiment (GRACE), we verify that the largest mass increases are tied to the southerly regime with cyclones west

of Greenland. For occurrences of the strongest southerly pattern, GRACE indicates a net mass increase of 1.29 Gt in the ice sheet accumulation zone (above 2 km elevation) compared to the 1.20 Gt of snowfall observed by CloudSat. This good overall agreement suggests that the analytical approach presented here can be used to directly quantify snowfall mass contributions and their most significant drivers spatially across the GrIS. While previous research has implicated this same southerly regime in ablation processes during summer, this paper shows that ablation mass loss in this circulation regime is nearly an order of

magnitude larger than the mass gain from associated snowfall. For daily occurrences of the southerly circulation regime, a mass loss of approximately 11 Gt is observed across the ice sheet despite snowfall mass input exceeding one gigatonne. By analyzing the spatial variability of snowfall and mass changes, this research provides new insight into connections between regional atmospheric circulation and GrIS mass balance.



## 1 Introduction

The Greenland ice sheet (GrIS) is currently the leading cause of global sea level rise, contributing an average increase of 0.47 mm each year in recent years (Shepherd et al., 2012; Van Den Broeke et al., 2016). Because of this contribution to sea level rise, GrIS mass balance plays an important role in the earth-climate system. While mass loss can result from a range of complex processes, snowfall is the major source of GrIS mass gain and is the largest control on inter-annual variability of GrIS mass balance (van den Broeke et al., 2009). Yet, because measurements are difficult to obtain, snowfall over the GrIS is poorly constrained and our understanding of snowfall's contribution to mass balance remains limited and uncertain (Noël et al., 2015; Vernon et al., 2013).

Previous studies have utilized models to show that mass input via snowfall over the GrIS is closely tied to atmospheric dynamics, with atmospheric circulation being the primary contributor to snowfall and accumulation variability (Noël et al., 2015; Brunswick et al., 2016). In terms of both snowfall mass input and ablation, research indicates that the GrIS is particularly sensitive to atmospheric variability on synoptic timescales (Schuenemann and Cassano, 2010; Doyle et al., 2015; Neff et al., 2014; Mattingly et al., 2018; Gallagher et al., 2020). Although these event-based studies have greatly improved our understanding of the relationship between the atmosphere and GrIS mass balance, few studies have focused on the impact of such events on snowfall. Important steps in this direction have been taken at specific Greenland in situ observation sites (Pedersen et al., 2018; Pettersen et al., 2018), but as of yet the connections between observations of spatial variation in snowfall and atmospheric circulation has not been examined. This analysis intends to broaden our understanding in this area, by relating atmospheric circulation to its impact on snowfall and changes in surface mass balance.

Recently researchers have begun to analyze Cloudsat derived estimates of snowfall across the Arctic, with Bennartz et al. (2019) studying the spatial and temporal variability of snowfall over Greenland and research by Edel et al. (2020) contextualizing snowfall across the broader Arctic climate. Work from McIlhattan et al. (2020) takes this a step further by analyzing detailed cloud properties and their impact on snowfall, but stops short of connecting these properties to atmospheric circulation. Thus, while these studies have provided important insight into Arctic snowfall, the work presented here advances this line of research by directly connecting the daily variability of atmospheric circulation to snowfall and its impact on Greenland ice sheet mass balance. Specifically, we relate the spatial and temporal variability of snowfall across the GrIS to the daily variability of regional atmospheric circulation regimes. By utilizing novel data products from CloudSat spaceborne radar observations of snowfall (Wood and L'Ecuyer, 2021) and categorizing local circulation patterns using self-organizing maps (SOMs), we assess the impact of the atmosphere on snowfall spatial variability across the GrIS and quantify snowfall mass input to the GrIS. Finally, by incorporating observations from the Gravity Recovery and Climate Experiment (GRACE), the snowfall mass input is more broadly contextualized to reveal the relationship between snowfall, atmospheric circulation, and daily GrIS mass changes. The work presented here is a significant step beyond previous research looking at average seasonal surface mass balance changes in the context of North American Oscillation, primarily because of the daily temporal resolution resulting from the unique methodology.



## 2 Observations

### 2.1 CloudSat snow product

The CloudSat spaceborne Cloud Profiling Radar (CPR) (Im et al., 2005; Stephens et al., 2008, 2018), a single-frequency 94 GHz W-band radar, provides the only snowfall observations available across the entirety of the GrIS. CloudSat orbits at an inclination of 98.2 degrees and the CPR has a spatial resolution of 1.4x1.8 km. CloudSat data used in this analysis were taken from the 5th release of the 2C-SNOW-PROFILE data product. This product uses a Bayesian estimation retrieval algorithm to estimate vertical properties of snowfall from reflectivity profiles measured by the CPR (Wood et al., 2013, 2014). CloudSat

observations from June $1^{st}$ 2006 to April $16^{th}$ 2011 were used here. Because this analysis requires uniform data to avoid assimilating diurnal biases tied to seasonality in the Arctic, utilizing observations from the period after the April 2011 CloudSat battery anomaly was not possible.

Previous research has determined CloudSat to be well suited for observing the light snow common at high latitudes such as over the GrIS (Cao et al., 2014; Norin et al., 2015; Palerme et al., 2017). Because this study is concerned with the impact of

snowfall on GrIS surface mass balance, snowfall rates are used from the 2C-SNOW-PROFILE product. Snowfall rates near the surface are not available due to ground-clutter interfering with the CPR; thus, these snowfall rates are derived from the fourth bin above the land surface. This is the first bin not contaminated by ground-clutter typically located roughly one kilometer above the Earth's surface. This gap is known as the snowfall 'blind zone' and it results in an approximately 10% underestimation of snowfall amounts in polar regions (Maahn et al., 2014). Despite this blind zone, research has shown that CloudSat observations

are closely correlated with mass accumulation at the GrIS surface and they are currently the best available spatial measurements of snowfall (Bennartz et al., 2019).

While CloudSat can potentially saturate during heavy snowfall, snowfall events of this magnitude are rare and CloudSat is particularly sensitive to the light snowfall common in the Arctic (Skofronick-jackson et al., 2014). The snowfall rates presented here are from above the ground-clutter zone and are assumed to statistically represent the snowfall at the GrIS surface. CloudSat

snowfall observations are provided in units of mm/hr but when used here the resulting snowfall averages are assumed to statistically represent the mean snowfall on the day of observation. Thus figures in this paper use units of mm/day when plotting snowfall. All observations were utilized in the form of a 1°x1°gridded data product.

### 2.2 GRACE, the Gravity Recovery and Climate Experiment

To contextualize the impact of snowfall on GrIS mass balance, this analysis compares snowfall mass input to mass balance

changes observed by GRACE. GRACE observations provide a unique continental perspective on mass balance changes and the NASA GSFC (Goddard Space Flight Center) GRACE data product provides the highest spatial resolution observations of GrIS mass changes currently of the various available GRACE data products (Luthcke et al., 2013). While there are other observations of GrIS mass balance, altimetry and interferometry approaches lack the equivalent spatial and temporal coverage provided by GRACE that is necessary to relate snowfall to mass changes. The GSFC GRACE observations are provided as a

type of pixel called "mass concentrations" cells, or simply mascons. Mass change data for the GSFC product is derived from





the level-1B GRACE observations directly and using an iterative method specifically designed for the retrieval of mass changes over the world's ice sheets. These GSFC GRACE data are provided as monthly mass changes for equal area mascons spanning nearly 15 years from 2003 to 2015.

For Greenland specifically, the ablation and accumulation regions of the ice sheet provide a boundary on the covariance of
GRACE observations based on the differences in the physical processes occurring in each region. Thus, mascons above and below two kilometers are constrained in the iterative GSFC solutions to be uncorrelated, significantly reducing signal leakage across this boundary and improving accuracy of the resulting product. This constraint separates mass changes caused by ablation and accumulation processes and provides an ideal product for studying GrIS mass changes in the context of snowfall. Although the GSFC data is provided for 1 arc-degree mascon cells, here mascons are aggregated into mass changes for the
entire ablation and accumulation regions of the GrIS. By aggregating data across these large regions, mass changes are being utilized near the native GRACE resolution of 300 km. In the context of this analysis, snowfall can then be attributed to mass changes above two kilometers, in the accumulation, region and thus relationships between snowfall and GRACE observations of GrIS mass balance can be determined.

To relate mass changes observed by GRACE to snowfall mass input, GRACE data were processed to quantify changes
in mass from month to month. GRACE data in their raw form are provided as observations of monthly deviations from the long term mean for each given mascon. To calculate mass changes from month to month, we have computed the gradient of the standard GSFC GRACE product using a simple central difference calculation for each month in the GRACE mascon time series'. This produces a new time series quantifying the mass change from one month to the next for each mascon. The data resulting from these GRACE gradient calculations will be referred to herein as "monthly mass deltas" ($\delta[mass]$). These
mass deltas are used to relate CloudSat observations of snowfall mass to the monthly variability of GrIS mass balance. In this analysis, individual mascon mass deltas are summed together to quantify regional changes in mass for three areas of the GrIS: above two kilometers, below two kilometers, and the entire GrIS. Thus mass delta refers to the net mass change of a region for a given month, as observed by GRACE. These data were utilized for the entire period of data available from the GSFC data product, from January 2003 to July of 2016.

**3  Methods**

In order to use these observations to assess atmospheric variability and its connection to snowfall, several techniques were employed. First, regions of Greenland where snowfall covaries in time were identified. Then, local atmospheric circulation was classified and a set of patterns representing the common local circulation states were established. Finally, combining the annual variability for identified snowfall regions and the categories of atmospheric circulation, spatiotemporal snowfall anomalies were
attributed to specific circulation patterns. All together, this methodology provides a foundation for using CloudSat observations to quantify the contribution from distinct atmospheric circulation patterns to GrIS snowfall mass input.





### 3.1 Snowfall regions and annual cycles

Because of the important role that the spatial distribution of snowfall plays in GrIS mass balance (Box et al., 2006), the first task is to characterize the spatial variability in snowfall across Greenland. To accomplish this, previous research has analyzed
CloudSat snowfall observations using broad temporal averages, most recently in Bennartz et al. (2019). Here, a complementary approach is taken by analyzing the contribution of atmospheric circulation to the spatial distribution of snowfall and thus bringing further understanding to these previous analyses.

To accomplish this, eight conglomerate regions were identified across Greenland using cluster analysis. Depicted in Fig. 1c, these regions represent areas where the daily variability of snowfall is considered 'similar'. These regions of similar snowfall
were identified by grouping $1°\text{x}1°$ CloudSat pixel observations by their temporal covariance. Thus, each region is a group of CloudSat pixels where snowfall covaries similarly in time, or put simply: if it's snowing in one pixel of the region, it is to some degree likely to be snowing in the adjoined pixels within the region. A total of eight regions were chosen to provide a minimum of 90% daily sampling coverage from CloudSat overpassess for each region. In testing, a larger number of regions provided insufficient data for the reconstruction of annual snowfall variability.

These regions were identified using a machine learning clustering algorithm. For this purpose, a covariance matrix was calculated for all pairs of CloudSat pixels using each pixel's time series of observations. This matrix was then weighted by the annual variability for each CloudSat pixel to ensure that the resulting clusters would create regions with similar annual cycles containing pixels whose time series covaried in time. These weighted covariance matrices were then used as input to a clustering algorithm. One could consider this the upscaling and grouping of pixels, similar to methods described in (Crane
and Hewitson, 2003). Clusters were constrained to only contain neighboring pixels, such that groups of pixels must remain geographically connected. The spectral clustering algorithm from the scikit-learn machine learning library (Pedregosa et al., 2011) resulted in the lowest internal cluster variance of all algorithms tested and was thus used to identify the regions presented here.

Across each region of clustered pixels in Fig. 1c, daily observations of snowfall from CloudSat were aggregated and the
resulting annual cycle associated with each region is shown in Figs. 1a and 1b. These annual cycles were derived by aggregating observations for all pixels in each region into a regional snowfall time series and Fourier decomposition was then used to find components of these time series with variability on timescales greater than two weeks (Gallagher et al., 2018). Using the identified low-frequency components, the annual cycles in Figs. 1a and 1b were constructed to determine the annual snowfall variability outside of high-frequency synoptic variability for each region.

These identified regions and annual snowfall cycles serve two distinct purposes in this analysis. The first purpose is to characterize unique features in the spatial and annual variability of snowfall across the GrIS. The second purpose is to compute daily snowfall anomalies as deviations from the expected intra-annual variability for each region. An observation of snowfall on a given day in a specific region of the ice sheet can be quantified as an anomaly relative to the expected value for that day of observation in the annual cycle as shown in Fig. 1. The word anomaly as used in this paper refers to snowfall observations
quantified relative to the spatiotemporal values shown in Fig. 1.



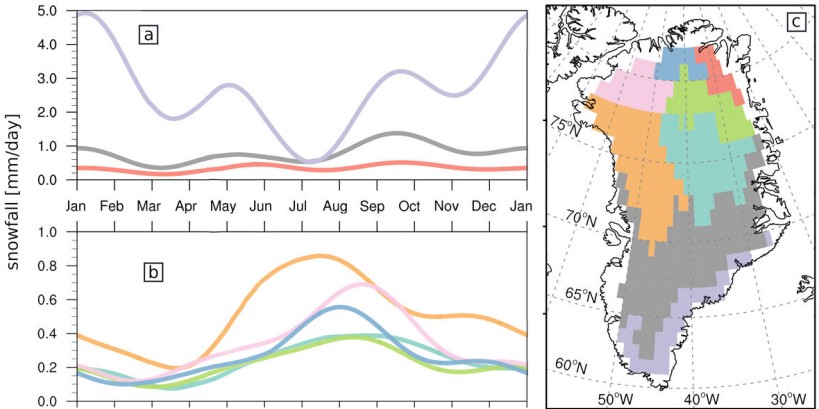

**Figure 1.** (a,b) The annual cycle of snowfall derived by using Fourier decomposition to deconstruct the temporal variability in snowfall observed by CloudSat for regions of similar variability presented to right, in units of mm/day. The top plot includes the three regions with a summer minima in snowfall while the bottom includes those with summer maxima. (c) Colored clusters indicate regions comprised of CloudSat pixel where daily snowfall covaries, as classified by machine learning clustering algorithm.

### 3.2 Categorizing atmospheric circulation

To study the variability in the spatial distribution of snowfall with respect to atmospheric circulation, the prominent circulation patterns influencing the GrIS spatial domain conditions have been identified (Fig. 2). This is because, for the broader GrIS, the spatial snowfall variability is determined primarily by the local atmospheric state (Schuenemann and Cassano, 2010; Pettersen
et al., 2018). By classifying atmospheric circulation, snowfall observations can be related to the broader atmospheric state influencing the spatial variability and magnitude of snowfall. In Fig. 2, grey shading and contours outside the Greenland sub-continent show the SLP circulation state while the color gradient contours within the Greenland subcontinent show the associated snowfall anomalies, described in Section 4.1.

For this analysis, these local circulation patterns were classified by the SOM algorithm (Hewitson and Crane, 2002; Kohonen,
2013) using daily sea level pressure (SLP) data from the National Centers for Environmental Prediction and National Center for Atmospheric Research (NCEP/NCAR) reanalysis (Kalnay et al., 1996) beginning January $1^{st}$ 1948 and ending December $31^{st}$ 2017. SOM classification uses an iterative nonlinear neural network algorithm to reduce the range of observed daily atmospheric states to a small group of representative categories. While other methods are available, SOMs are particularly well suited for the task. This is because the SOM algorithm minimizes the number of human assumptions about the underlying
structure of the data and results in a simple grid of circulation states, where similar circulation states are neighbors in the map. The SOM algorithm's ability to produce an accurate and complete map of daily circulation states in a format that is simple to interpret makes it a common tool for categorizing circulation states (Sheridan and Lee, 2011).

Using the NCEP/NCAR SLP data, days identified by the SOM algorithm as having similar SLP fields are grouped together under a representative circulation pattern, called "nodes" in SOM parlance. Observations can then be related to regional cir-
culation state by aggregating daily data corresponding to the days associated with each identified circulation pattern. Before





it was utilized, the gridded NCEP/NCAR data product was modified in the following ways. Terrain above 1,000 meters were masked to remove SLP values significantly detached from surface observations. The remaining data was interpolated from the native 2.5° grid spacing to an equal-area grid of 50 km, such that northerly and southerly latitudes are given equal weighting by the SOM algorithm. For each day the domain mean SLP was subtracted from the gridded SLP values to capture the circulation

anomalous to the mean background state. Finally, these processed data were then provided to the SOM algorithm as inputs.

Here, a SOM classification of 20 circulation patterns was chosen. A total of 20 circulation patterns were chosen to strike a balance between identifying the complete set of unique region patterns while maintaining a sufficient sample of observations related to each circulation pattern (Gallagher et al., 2018). Other previous analyses have also used SOMs to categorize Greenland's local atmospheric circulation and the SOM classification used in this analysis is consistent with these prior SOM

climatologies (Schuenemann and Cassano, 2010; Mioduszewski et al., 2016; Gallagher et al., 2018). The final outcome of this process is a map of daily atmospheric circulation patterns (Fig. 2), organized by similarity, representing the common local SLP circulation patterns. For each identified circulation pattern, the SOM algorithm provides a list of days that the pattern occurred. Thus, observations can be related to the SOM by aggregating observations for the list of days for each circulation pattern.

### 3.3 Snowfall anomalies

For sparse CloudSat observations of snowfall, spatiotemporal snowfall anomalies are calculated as the deviation of a particular observation from the annual cycle for the region where the observation was made (Fig. 1). These anomalies are then averaged for all observations associated with a given SOM circulation state and spatial maps are aggregated to quantify the average snowfall anomaly attributed to each circulation pattern. The motivation for calculating snowfall anomalies is to de-convolve the relationship between changes in snowfall on inter-annual timescales with snowfall variability attributed to atmospheric

circulation on daily timescales. As the occurrence frequency of a specific circulation pattern varies throughout the year (Fig. 3), observations attributed to a given circulation state must be anomalous with respect to the snowfall expected at the time of the occurrence. A prior implementation and discussion of the methods used here can be found in Gallagher et al. (2020).

Using the anomalous snowfall values described in Section 3.1, snowfall observations are attributed to the SOM categorized circulation states in Fig. 2. The average anomalous snowfall attributed to each circulation pattern in Fig. 2 reveals a strong re-

lationship between the snowfall variability across the GrIS and atmospheric circulation state, showing that anomalous snowfall events in a given region of the GrIS are dependent on the circulation state. According to Fig. 2, southerly transport events relate to broad increased snowfall across much of the GrIS, while most other atmospheric circulation conditions relate to more localized snowfall in regions with upslope flow. In particular, zonal transport regimes surrounding [c,3] relate to highly localized but large increases in snowfall. While these are generalizations, a detailed discussion of these relationships follows in Section

205 4.1.





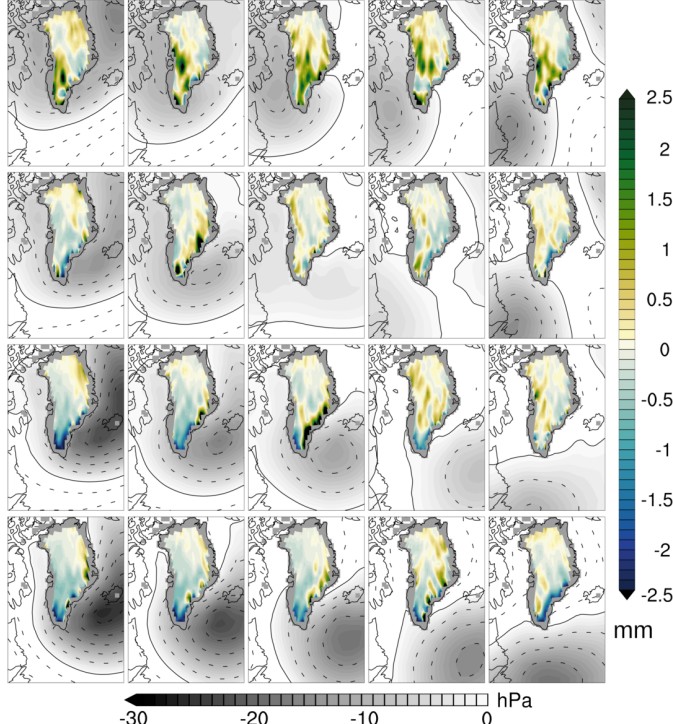

**Figure 2.** Snowfall anomalies over the Greenland attributed to common local atmospheric circulation patterns. Snowfall anomaly is relative to the region and annual cycle as identified in Fig. 1, with snowfall from CloudSat observations in units of mm/day. Local atmospheric circulation patterns were identified using the SOM algorithm and the low-pressure centers for each circulation pattern are plotted in grey.

To accurately interpret regional circulation patterns in Fig. 2 and their relationship to observations, it is important to understand how the frequency of each pattern changes throughout the year. Here, Fig. 3 shows that the likelihood of occurrence varies intra-annually for each circulation pattern. Fig. 3 reveals some general patterns in the annual cycle for the various regional circulation regimes. Southerly transport patterns surrounding node [c,1] occur primarily in summer, with only occasional

winter occurrences. The strongest northerly transport patterns surrounding [a,4] occur primarily in winter, with very near zero occurrences during summer months. In contrast, zonal circulation patterns around node [c,3] occur throughout the year. Some individual zonal patterns are more likely to occur in winter or summer dependent on cyclone strength and position. The intra-annual variability shown here is consistent with prior regional climatologies (Schuenemann and Cassano, 2010; Mioduszewski et al., 2016).



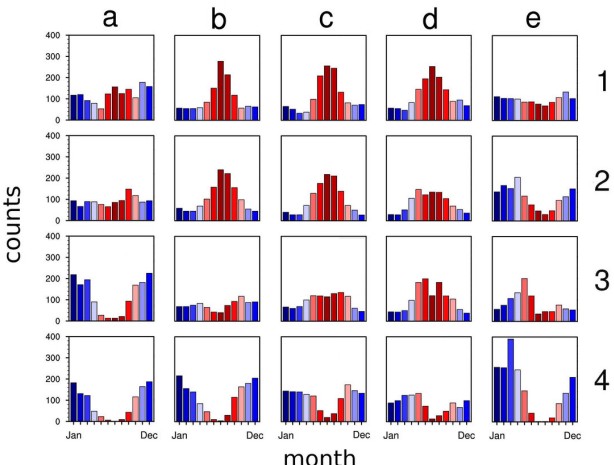

**Figure 3.** The annual distribution of occurrences of each circulation pattern in Fig. 2 binned by months. Occurrences are counted for all days in the NCEP/NCAR reanalysis from 1948 through 2017. Coloring is used only to visually differentiate months.

## 4 Results

### 4.1 Regional circulation and snowfall variability

Regional snowfall clusters presented in Fig. 1 establish the spatial variability in snowfall across the GrIS and also provide the basis for attributing daily snowfall anomalies to atmospheric circulation. Several notable features of snowfall over Greenland are presented in Fig. 1. The largest snowfall occurs over the southeastern portion of the ice sheet near the coast, with snowfall amounts roughly five times more than any other region of the GrIS. While all other regions experience snowfall maxima of approximately 1.0 mm/day in summer or late fall, this southeasterly coastal region reaches a 5.0 mm/day maximum snowfall of in January. For regions north of 75° latitude, snowfall annual cycle maxima do not exceed 1 mm/day. Of regions north of 75°, the western coastal regions shown in pink and orange have the highest peak snowfall while regions in the northeast have the lowest.

Relative to these annual cycles, Fig. 2 attributes snowfall anomalies to the daily variability of regional circulation. Fig. 2 demonstrates the important role of atmospheric circulation in the daily spatial variability of snowfall, showing that each region of the GrIS is unique in its relationship to atmospheric circulation.

The southeastern region, a region with a winter snowfall maximum, has the largest anomalous snowfall when cyclone centers are located very near to the southeastern coast. These circulation patterns relate to easterly transport and strong upslope flow against the steep topography of the southeast coast and thus large anomalies tied to orographic forcing. The magnitude of anomalous snowfall in the southeast caused by these easterly patterns is highly dependent on the exact position of the low-pressure center. Circulation patterns [c,3] and [b,2], with cyclones nearest to Greenland's tip, cause the largest snowfall



anomalies in this southeastern region. Here, the interaction of this large-scale flow with local fjords, inlets, and peaks, leads to distinct enhanced and diminished snowfall dependent on geography.

Over the central GrIS, the largest positive snowfall anomalies relate to southerly transport patterns surrounding node [c,1]. Broad positive snowfall anomalies across the GrIS are seen for circulation patterns where the cyclone center is north of 55° latitude and west of the Greenland subcontinent over Baffin Bay. These southerly transport patterns have previously been related to moisture intrusions and atmospheric rivers (Liu and Barnes, 2015; Neff et al., 2014) and are quantified here for the first time in terms of their relationship to the spatial distribution of observed snowfall. For the most prominent of these southerly

patterns, node [c,1], anomalously high snowfall extends across the GrIS covering all but the northernmost coastal region.

     In regions north of 75° latitude on the GrIS, the snowfall anomalies associated with atmospheric circulation are the lowest of anywhere on the GrIS. Positive anomalies on the north and northeastern coast are seen for circulation patterns with cyclone centers above Iceland, in [a,3] and neighboring nodes. These circulation patterns correspond with northerly upslope flow from the Greenland Sea and Arctic Ocean, with snowfall anomalies significantly smaller in both magnitude and spatial extent when

compared to snowfall in other GrIS regions.

     In order to provide further context for the relationship of snowfall to daily circulation variability, the net snowfall mass input for each SLP circulation pattern was calculated. In this way, Fig. 4 reveals the cumulative snowfall mass input in gigatonnes for each circulation pattern. Unlike Fig. 2, these values are not calculated as anomalies but instead provide the total snowfall mass observed by CloudSat, averaged for each circulation pattern. Thus, Fig. 4 shows cumulative snowfall mass input to the

GrIS as described by the daily variability in atmospheric circulation. In contrast to the anomalies shown in Fig. 2, the word cumulative in this text refers to the total magnitude of snowfall for each circulation pattern.

     Motivated by the mechanics of GrIS mass changes in the ablation and accumulation zone, Fig. 4a aggregates the cumulative snowfall across the entirety of the GrIS while Fig. 4b includes only observations where surface elevation is above two kilometers, and Fig. 4c includes only snowfall observations where surface elevation is below two kilometers. Fig. 4a is thus the sum of

the values presented in Figs. 4b and 4c. An elevation of two kilometers was chosen to delineate the ablation and accumulation zones as previous research has shown that this elevation approximately demarcates the dry region of the GrIS, above which surface melt generally does not occur (McMillan et al., 2016; Hall et al., 2013). This also facilitates comparisons to the GSFC GRACE mass change data product. While the high-resolution spatial variability in snowfall is shown in detail in Fig. 2, Fig. 4 is meant to provide a cumulative value for these spatial characteristics. These quantities are best understood when studied

in the spatial context of maps presented in Fig. 2. Values are presented in gigatonnes to facilitate comparisons to the literature and align results presented here with their direct connection to mass balance and thus the fate of the GrIS.

     From Fig. 4, the circulation patterns responsible for the largest snowfall mass input to the Greenland ice sheet are southerly transport patterns. Node [c,1] in particular relates to the largest snowfall, with an average daily mass input of 1.66 Gt across the entirety of the GrIS. Fig. 4b shows that the majority of this snowfall occurs in the accumulation zone of the GrIS, with 1.2

Gt of the total 1.66 Gt of snowfall occurring over surfaces above two kilometers in elevation.

     In terms of cumulative snowfall mass input, zonal pattern [c,3] is near the magnitude of these southerly patterns, with a cumulative snowfall mass input of 1.47 Gt, only ten percent less than southerly pattern [c,1]. However, as shown by the spatial



snowfall anomalies in Fig. 2 and the cumulative mass input above two kilometers of 0.7 Gt from Fig. 4b, the majority of this snowfall occurs on or near the southeastern coast. Also, this particular zonal pattern stands alone in its snowfall impact, as the circulation patterns neighboring [c,3] result in significantly less snowfall mass input. The differences in cumulative snowfall mass input for southerly and zonal circulation patterns is most stark for snowfall above two kilometers. Although both southerly pattern [c,1] and zonal pattern [c,3] induce relatively large amounts of snowfall in total, [c,1] transports nearly double the snowfall mass to elevations above two kilometers.

Another noteworthy detail of these results is the snowfall impact of circulation pattern [a,1], a broad low pressure system spanning the GrIS. Fig. 4a shows that this pattern results in a significant amount of snowfall, a total of 1.52 Gt, which is near the magnitude of pattern [c,1]. However, the spatial distribution of this snowfall differs significantly, with anomalous snowfall increases on both the southwest and northeastern coasts of the GrIS. As with zonal circulation pattern [c,3], northerly transport pattern [a,1] also has a diminished impact on snowfall mass input above two kilometers.





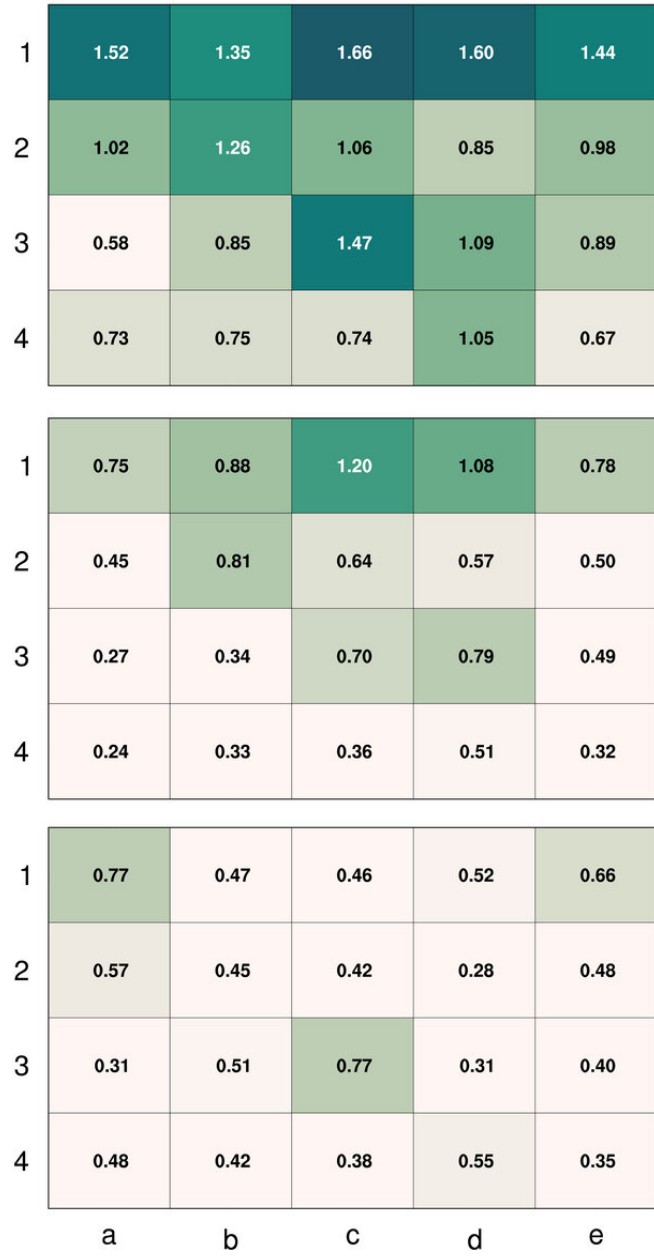

**Figure 4.** Top: The average cumulative snowfall mass across the entire ice sheet observed by CloudSat for each daily circulation pattern, in gigatonnes. Middle: Average total snowfall mass at elevations above two kilometers on the GrIS. Bottom: Average total snowfall mass at elevations below two kilometers on the GrIS. The coloring is used only to indicate qualitatively the relative difference in magnitude between patterns.





Quantifying snowfall magnitude and its spatial variability tied to atmospheric circulation provides important information

on relationships between the atmosphere, snowfall, and GrIS mass balance. Yet, snowfall is only the mass input to the GrIS. Previous research has examined how atmospheric variability relates to GrIS mass changes (Gallagher et al., 2020; Neff et al., 2014; Fettweis et al., 2011) and concluded that southerly transport, resulting moisture advection, and temperature increases on short timescales contribute significantly to GrIS ablation. These are the same southerly events revealed here to be responsible for the largest snowfall across Greenland. Thus, in the following sections, observations of mass changes are used to quantify

the balance between snowfall mass input and net mass changes of the GrIS for this influential southerly regime.

## 4.2 Circulation, snowfall, and GrIS mass balance changes

### 4.2.1 Comparing GrIS mass changes from GRACE to CloudSat snowfall

The goal of incorporating observations from GRACE into this analysis is to provide a comparison between the impact of atmospheric circulation on snowfall mass input (Fig. 4) to the actual net mass changes of the GrIS. This is because previous

research has demonstrated that the southerly regime, related here to the largest daily snowfall (Fig. 4), corresponds to anomalously warm near-surface temperatures (Gallagher et al., 2020) and melt of the GrIS surface (Mioduszewski et al., 2016). Thus, this section compares GRACE mass changes to the daily snowfall mass variability in Fig. 4 to establish and quantify the relative balance between snowfall and net mass changes for this important southerly circulation regime. Quantifying the balance between southerly circulation, snowfall mass input, and GrIS mass changes in this way provides important insight into the net

impact of regional atmospheric variability on the GrIS.

Three figures are provided to examine these relationships. Fig. 5 establishes a baseline for comparing GRACE mass changes to the snowfall framework based on regional circulation constructed here. Fig. 6 then directly connects mass changes observed above two kilometers to pattern [c,1], the circulation pattern related to the largest snowfall. And finally, Fig. 7 improves on Fig. 6 by providing a more statistically robust comparison between occurrences of this important southerly regime and cumulative

mass changes of the entire GrIS. The following paragraphs discuss these figures in detail.

First, in Fig. 5, mass changes observed by GRACE are directly compared to total monthly snowfall mass values derived from the daily variability of snowfall presented in Fig. 4b. This is done to provide a baseline for comparing the two independent data sources. In Fig. 5, the total snowfall for a given month (x-axis) was constructed by summing the mean snowfall values above two kilometers (shown in Fig. 4b) for all circulation patterns occurring in that month. For example, if in July pattern [c,1]

occurs 10 times, this would account for a snowfall above two kilometers of 12 Gt. Then, if on the remaining 21 days of the month low-snowfall pattern [e,1] occurred averaging snowfall of 0.5 Gt each day, the cumulative snowfall for this July would be 1.2*10+0.5*21=22.5 Gt.

Thus the total snowfall mass in a given month can be plotted against the mass change of that month observed by GRACE. In Fig. 5 this comparison is made exclusively in the accumulation zone above two kilometers, because, above two kilometers,

snowfall is the most significant component of mass changes. The other primary component, changes in ice sheet discharge, varies around a mean of 50 Gt/year (King et al., 2018; Joughin et al., 2018) in the accumulation zone. Variability of ice sheet



discharge on monthly timescales is significantly less than the magnitude of snowfall mass input presented here in Fig. 4 (King et al., 2018). Thus, snowfall is the primary driver of GrIS mass changes above two kilometers and circulation patterns related to heavy snowfall can be reliably connected to observations of mass changes in the accumulation region of the GrIS.

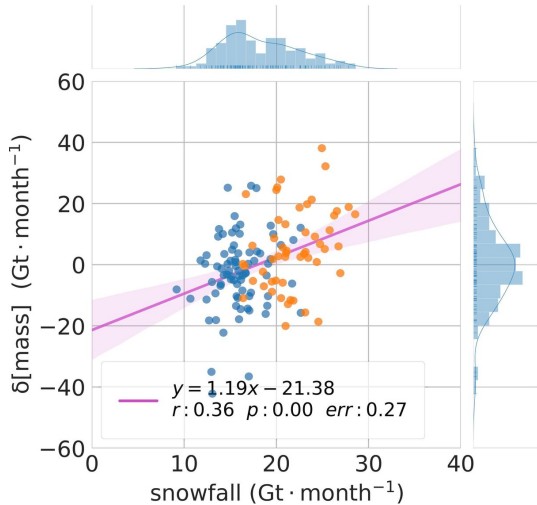

**Figure 5.** A linear regression between the cumulative change in mass above two kilometers for a given month as seen by GRACE to the total snowfall mass values summed together for all circulation patterns occurring in that month using daily snowfall mass values related to atmospheric circulation from Fig. 4b. These values are for only the accumulation region of the GrIS above two kilometers where snowfall is the primary control on changes in mass. The regression line includes 95% confidence intervals in transparent red. The regression slope indicates that for every 1.19 Gt of mass change seen by GRACE in a given month there is 1 Gt of snowfall accounted for in Fig. 4b, although with significant variance for individual months. Each dot is colored by the time of year that it occurred, with melt months shown in orange and defined to be May, June, July, August, and September

Fig. 5 shows that for every 1 Gt of snowfall observed by CloudSat, a corresponding mass change of 1.19 Gt of is observed by GRACE in a given month. Despite the significance of this linear regression (p=0), an r value of 0.36 indicates a large amount of variance between the two parameters. While a regression slope of 1.19 is almost 20% above a perfect correspondence between observations of snowfall and mass changes, a slope of 1.0 is within the 95% confidence intervals of the regression as a result of the significant variance. This is consistent with other studies finding CloudSat underestimates high-latitude snowfall amounts (Bennartz et al., 2019; Maahn et al., 2014). With consideration for the variance, Fig. 5 demonstrates that the snowfall mass attributed to circulation patterns in Fig. 4b corresponds roughly to mass changes observed by GRACE on monthly timescales.

An important feature of the results in Fig. 5 is the impact of ice sheet dynamics on the intercept of the regression line. While snowfall is the major mass contributor to the GrIS, above two kilometers ice sheet dynamics are completely responsible for mass losses. Thus, ice sheet dynamics is a confounding variable in the linear regressions between snowfall and mass changes presented here. To understand this more clearly, projecting the trend from Fig. 5 indicates that in the unrealistic scenario of a month with zero snowfall ice sheet discharge would result in a mass loss of approximately 21 Gt in the region of the GrIS



above two kilometers. Also, looking at the confidence intervals on this projection, the bounds on dynamic mass loss above 2km is likely between 10 and 30 Gt in any given year with extreme years further outside these bounds. In this way, the inter-annual variability of ice sheet dynamics likely explains the variability in agreement between individual months in the comparison
between observations of snowfall from CloudSat and mass changes from GRACE.

The variability due to ice sheet dynamics shown here in Fig. 5 mirrors results presented in previous studies. King et al. (2018) showed that dynamic discharge of the northwestern GrIS accounts for an average mass loss of 205 Gt per year while Joughin et al. (2018) shows that ice velocities in this region average approximately 100 meters per year. Joughin et al. (2018) also shows that ice velocities on the central GrIS above two kilometers averages approximately 25 meters per year, roughly
25% of the coastal flow velocity. According to these combined studies, mass discharge on the central GrIS would be on the order of 50 Gt per year using these approximations with inter-annual variability dependent on changes in flow rate. This back of the envelope comparison shows that the comparison presented is physically realistic to first order. However, an accurate estimate of central GrIS flow in terms of mass could not be found in the recent literature. A detailed model of the ice sheet response to snowfall would be required to more deeply explore these results and the variance related to dynamics. With these
considerations, Fig. 5 establishes a baseline to compare mass changes observed by GRACE to snowfall mass input derived from CloudSat observations.

### 4.2.2    Characterizing mass changes in the 'active' southerly circulation regime

Previously cyclones west of Greenland occurring primarily in summer were related to the largest snowfall mass input from CloudSat observations across GrIS. Here this southerly regime is investigated in the full context of GrIS mass changes, in-
cluding in the ablation region, exploring both mass increases from snowfall as well as charactizing the coinciding mass losses in the ablation region. Because of the findings showing extreme snowfall coinciding with extreme mass loss, this southerly circulation regime is designated the 'active' Greenland circulation regime.

Thus, Fig. 6 plots the number of monthly occurrences of circulation pattern [c,1] against the monthly mass deltas observed by GRACE above two kilometers observed for that month. Because of the monthly timescale of GRACE observations, mass
deltas could not simply be assigned to the daily circulation patterns in Fig. 2. Instead, the relationship between mass changes for each month and the number of occurrences of these southerly circulation patterns are quantified by a linear regression. Accordingly, the linear regression slope indicates the change in mass relative to the number of occurrences of these southerly circulation patterns in a given month. The regressions presented in Fig. 7 were plotted for all circulation patterns in Fig. 2, both individually as well as in groups. The largest regression slope was seen for [c,1], in agreement with the maximum
cumulative snowfall above two kilometers observed by CloudSat. This approach allows for a direct comparison to CloudSat snow accumulations for circulation patterns presented previously in Fig. 4.

Fig. 6 shows a mass gain in the accumulation region of 1.29 Gt for each occurrence of circulation pattern [c,1], although with significant variability around the regression. While the slope of this regression (1.29 Gt/occurrence) is similar in magnitude to the daily contribution of [c,1] to snowfall mass input in Fig. 4b (1.20 Gt), the relationship is not perfect. A positive slope is
only narrowly included in the 95% confidence interval of the regression and the p and r values indicate significant variability





not constrained by these two parameters. There are three factors that contribute to the significant variance for this regression: 1) natural intra-pattern variability in the snowfall for occurrences of each circulation pattern 2) variability in the circulation patterns occurring in a months that are not included in the regression 3) variability in ice sheet dynamics and outflow. Despite this variance, this regression provides further support for the importance of [c,1] to snowfall mass input.

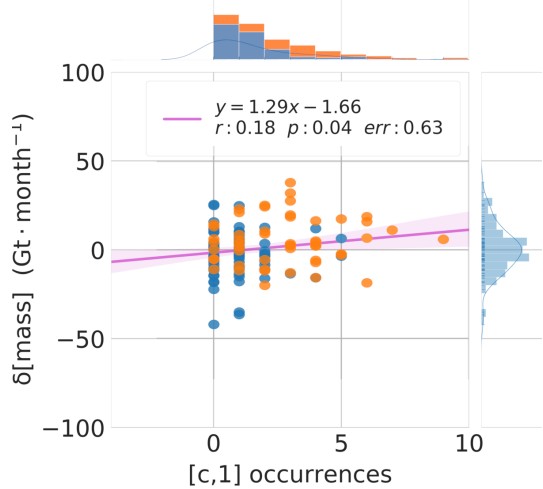

**Figure 6.** A linear regression between the monthly frequency for all occurrences of southerly pattern [c,1] and the cumulative change in mass above 2km elevation for all months as seen by GRACE. The regression line includes 95% confidence intervals. Each dot is colored by the time of year that it occurred, with melt months shown in orange defined to be May, June, July, August, and September. The regression line includes 95% confidence intervals.

Because of the relatively weak statistical correspondence between mass changes and occurrences of pattern [c,1] alone, in Fig. 7 we have combined the three circulation patterns with the largest snowfall. This improves statistics beyond Fig. 6 by sampling a larger number of days in each month. In Fig. 7, mass deltas from GRACE observations are plotted against the number of occurrences of southerly transport patterns [b,1], [c,1], and [d,1] for each month. These plots show monthly GrIS mass changes for all mascons above two kilometers (Fig. 7a), below two kilometers (Fig. 7b), and for the entire GrIS (Fig. 7c).

Comparing mass changes in the accumulation zone from Fig. 7a with snowfall in Fig. 4b, the snowfall mass input of the three patterns averages 1.05 Gt while a cumulative mass change of 0.96 Gt is observed by GRACE for each occurrence of these three patterns, with some variance. This shows, within the significant measurement uncertainties, that GRACE observations correspond with the mass changes caused by snowfall in the accumulation zone.


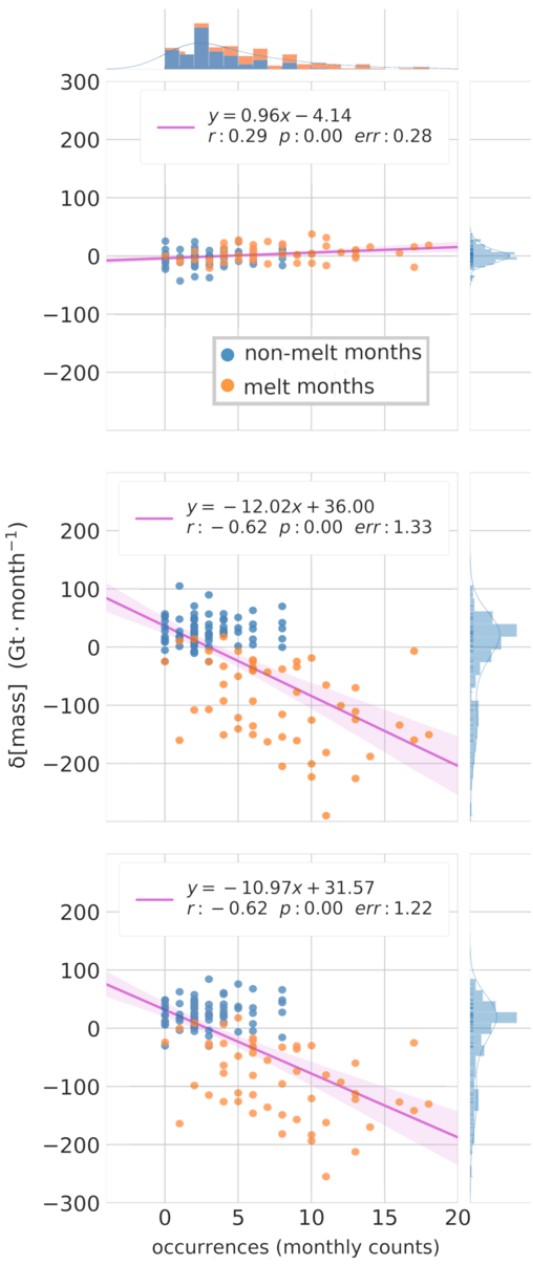

**Figure 7.** A linear regression, as in Fig. 6, but for all occurrences of southerly patterns [b,1], [c,1] and [d,1]. The regression line includes 95% confidence intervals. Each dot is colored by the time of year that it occurred, with melt months defined to be May, June, July, August, and September. Top: mass change for GRACE mascons above 2 km. Middle: mass change for GRACE mascons below 2 km. Bottom: mass change for all Greenland GRACE mascons, both above and below 2 km.





Comparing the mass changes in the ablation and accumulation regions of the ice sheet in Fig. 7, the relative magnitude of
mass loss and snowfall for these southerly events can be assessed. In Fig. 7a, the regression slope describes a relationship where
GrIS mass above two kilometers increases 0.96 gigatonnes for each occurrence of these southerly transport patterns, with a
similar relationship observed year round in this typically non-melting region. Conversely, below two kilometers on the GrIS,
the linear regression for all months in Fig. 7b indicates that mass decreases 12.02 gigatonnes for each additional occurrence
of these southerly patterns. It is worth noting that when considering only the non-melt months, linear regression indicates that
mass increases by 2.2 gigatonnes for each additional occurrence of the southerly patterns, providing a reasonable sanity check
on this methodology. When considering the whole GrIS over all months (Fig. 7c), coastal ablation is an order of magnitude
larger than snowfall mass increases, leading to a net mass loss of 10.97 Gt across the entire ice sheet for each occurrence of
southerly circulation patterns, showing that for these patterns mass loss from ablation is significantly larger than snowfall mass
input. Again, for only the winter months over the full GrIS, the methodology suggests a net mass increase of 2.2 gigatonnes per
occurrence of southerly patterns. Combined together, Figs. 5, 6, and 7 are intended to support the discussion and conclusions
based on the relative proportions of snowfall mass input and GrIS ablation as well as provide a baseline for understanding
snowfall in the broader context of GrIS mass changes.

To put this mass loss from ablation in the context of the atmospheric circulation framework used here, NCEP/NCAR reanaly-
sis daily mean temperatures were related to the circulation patterns shown in Fig. 2. Fig. 8 shows the mean daily position of the
zero degree isotherm for occurrences of each circulation pattern. Although the zero degree isotherm position is dependent on
the seasonal fraction of occurrence of each pattern, it provides a simple proxy to understand how circulation patterns presented
here relate to GrIS ablation processes. The discussion of ablation here is supported by a more detailed analysis of temperature,
cloud impacts, and atmospheric circulation patterns found in Gallagher et al. (2020), as well as other literature detailing the
atmospheric contribution to ablation processes (Neff et al., 2014; Hanna et al., 2014; Fettweis et al., 2011; Mioduszewski et al.,
395   2016).

The relationship between atmospheric circulation and warm temperatures on the GrIS in Fig. 8 illustrates the connection
between southerly transport and broad warming. Southerly circulation patterns surrounding node [c,1] relate to the largest
fractions of the GrIS with temperatures above 0°C. In the context of the snowfall results presented at the beginning of this
paper, these southerly transport patterns are thus the most 'active' atmospheric regimes in terms of their impact on daily
variability in GrIS mass balance. Node [c,1] in particular relates to the largest daily snowfall mass input of any circulation
pattern at 1.66 Gt as well as the largest broad warming above 0°C, and resulting melt as supported by Mioduszewski et al.
(2016). However, the relative magnitude of these co-occurring and competing impacts on mass balance had not previously
been quantified. In this way, Fig. 7 provides an estimate of the balance between ablation and accumulation, showing that mass
loss in the ablation zone is significantly larger than snowfall for occurrences of this southerly regime.





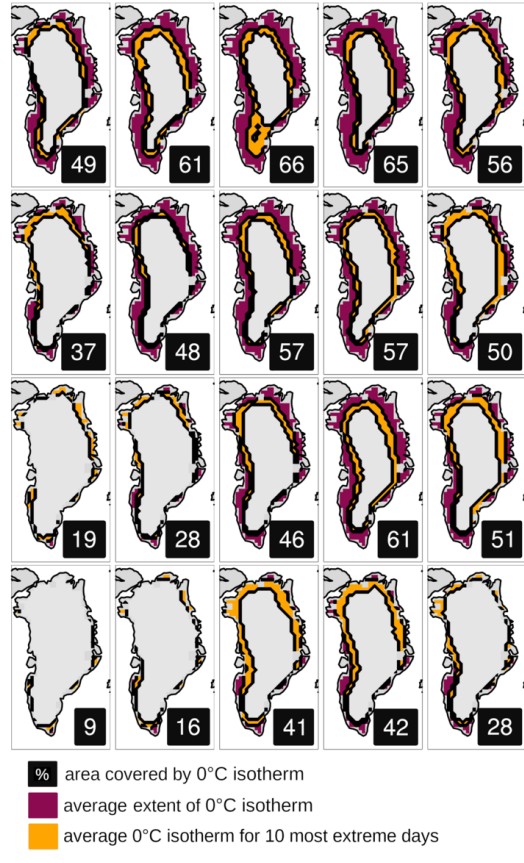

**Figure 8.** The average position of the zero degree isotherm for all days associated with each circulation pattern. Data are from gridded NCEP/NCAR reanalysis fields of two meter temperatures for the entire NCEP/NCAR reanalysis record from 1948 through 2017. Red coloring shows the mean position of the zero degree isotherm for all days for the full set of reanalysis data. Orange coloring indicates the position of the zero degree isotherm for only the 10 warmest days during the CloudSat data period from 2006-2011. The numbers in the black box indicate the percentage of the total area of the GrIS covered by the orange region for these 10 most extreme days.

While a large portion of this paper has focused on the atmospheric drivers of large snowfall events and their relationship to ablation these events occur on relatively short timescales, most often in summer. In the framework of this paper, other relationships between snowfall and atmospheric circulation can also be discussed. In opposition to these southerly events most frequent in summer, these results show that circulation patterns surrounding [a,4] related to northerly transport occur most frequently in winter. Despite their nominal snowfall, these northerly patterns occur primarily in winter when GrIS ablation is at

a minimum and thus relate to consistent mass increases. In this way, these results show that moderate snowfall in winter relates to the largest mass increases on long-term timescales, in agreement with prior research (Koenig et al., 2016). Although the focus here is primarily on the so-called 'active' southerly circulation regime, the analysis, methodology, and figures provide





context for the complete variability in local atmospheric state. To keep the written discussion from being even more verbose further detailed understanding of these patterns may be gleaned from the graphical figures.

### 4.2.3    Discussion of uncertainty

While this work brings to light new understanding, assessing the potential uncertainty in the results is as important as it is difficult. The primary conclusions of the paper are motivated by the snowfall anomaly calculations made using CloudSat observations aggregated in spatial clusters and then processed via spectral analysis. Propagating the statistical uncertainty through this machinery would be Herculean in effort as well as difficult to interpret scientifically. Thus, in lieu of being able to directly quantify the resulting uncertainty, we will instead discuss the validity of using these observations as inputs for these methods.

While individual snowfall observations retrieved from CloudSat profiles have large uncertainties, observations averaged in time, have been shown to compare very accurately to surface in situ measurements of snowfall measurements over Greenland (Bennartz et al., 2019; Boening et al., 2012). In this way, this analysis uses the aggregated mean of a large sample of observations: the mean snowfall rate is sampled from many CloudSat profiles across a 1x1 degree pixel which is then averaged for the locally clustered region to create the seasonal background variability, which is in turn used to calculate the anomaly for the average of all snowfall observations for a given circulation pattern. Thus any particular anomaly value plotted in Fig. 2 is the average of many hundreds of snowfall observations, while the cumulative GrIS snowfall mass input values from Fig. 4 is the average of many thousands. The conclusion being that the significant uncertainty and noise in individual observations will be largely canceled by the processing. This is evidenced by the relatively smooth transitions in the spatial snowfall anomalies driven by circulation (Fig. 2), indicating that the pixel-by-pixel noise is significantly lower than the observation signal.

For the GRACE component of the analysis, much like CloudSat snowfall observations, the uncertainty of any individual mass delta observation for a particular mascon is very high, due primarily to the noise present in the GRACE measurement system (Luthcke et al., 2013). Further, previous research has also shown that for averaging mascon observations across large areas of the GrIS uncertainty contributions from noise are largely negligible (Ran et al., 2017). For the analysis presented here, monthly mass changes are averaged for two distinct large areas of the GrIS above and below two kilometers. As a result, the significant uncertainty resulting from noise in GRACE measurements is minimized for this analysis.

These paragraphs are the attempt of the authors to acknowledge there are many factors at play in presenting a comparison between such disparate datasets using relatively complex methodologies. Although the very close agreement in mass changes is extremely compelling, the analysis is most compelling for the fact that the results provide new detailed information about key GrIS processes leading to conclusions that can be scientifically reasoned with.

## 5    Summary and conclusions

By looking at CloudSat snowfall observations in a framework with atmospheric circulation, this paper constructs a spatial relationship between snowfall and the mass balance of the GrIS on daily timescales. To contextualize these results in terms





of GrIS mass balance more broadly, GRACE observations were utilized to show how snowfall mass input relates to net mass changes.

The snowfall observations presented here show that snow input peaks for the majority of the GrIS in the mid-to-late summer months. Further, summer snowfall is shown to be two to four times greater in magnitude than snowfall in winter, with the inter-annual variability in snowfall relating to the seasonal variability in occurrence of specific circulation patterns. Mapping

the spatial variability in snowfall to these regional circulation patterns shows that snowfall across Greenland relates closely to the position and strength of cyclonic systems, with localized snowfall closely tied to the location of onshore atmospheric flow. More specifically, the southerly transport regime with cyclones west of the GrIS, most commonly occurring in summer, is implicated in the largest snowfall both in mass input and spatial extent. More specifically, for the most prominent southerly circulation pattern, daily snowfall mass input across Greenland is a total of 1.66 Gt. Spatially, this southerly circulation regime

is the primary source of snowfall on the central ice sheet as, with 1.2 Gt of the total 1.66 Gt falling in the accumulation zone above two kilometers elevation.

Observations of mass changes from GRACE independently show that these same primarily-summer southerly circulation patterns also relate to significant mass loss in the ablation zone of the GrIS. Thus, the southerly regime has been designated the 'active' circulation regime due to the quantified relationships between large snowfall mass input coinciding with significant

mass loss. While the snowfall for this southerly regime averages between 1.3 and 1.6 Gt per day, GRACE observations implicate this southerly pattern in a mass loss on the order of 11 Gt per each daily occurrence.

More generally, across Greenland daily snowfall mass input varies between 0.58 and 1.66 Gt dependent the atmospheric circulation conditions for a particular day, with anomalously extreme days falling outside these bounds. Above two kilometers in the accumulation region there is even higher variability in daily snowfall mass input. In the accumulation zone the largest

snowfall results from southerly transport with a maximum of 1.20 Gt relating to daily occurrences of pattern [c,1]. This is in comparison to northerly events that contribute as little as 0.24 Gt of snowfall to the central ice sheet. These results provide a new constraint on the daily spatial variability of snowfall mass input across Greenland using direct observations of snowfall.

Because of the novel nature combined methodologies presented in this paper, a direct comparison between GRACE observations and the aggregated results of snowfall mass input is presented. Through this comparison it is shown that observations

from GRACE of mass change approximately agree in magnitude with the aggregated results from CloudSat snowfall. For the methodology presented here, every 1.19 Gt of snowfall mass input observed by CloudSat corresponded with a mass increase observed by GRACE of 1.0 Gt, an approximately 20% overestimation of snowfall mass input using aggregated CloudSat observations. While imperfect, the agreement between these independent observations provides compelling evidence in favor of the methodologies presented, particularly considering the disparate nature of CloudSat and GRACE observations. This disagree-

ment is also within the uncertainty of the observations and approximately near the magnitude of previously reported tendencies for CloudSat to over-estimate snowfall in the Arctic (Bennartz et al., 2019; Maahn et al., 2014).

The work presented in this paper provides new insight, rooted in observations, into the relationships between broader atmospheric conditions, spatial snowfall variability across Greenland, and corresponding GrIS mass changes. Further, because this novel methodology constructs never-before-seen maps of the daily spatial variability of snowfall across Greenland, the



authors hope that this work will lead to more accurate research of snowfall and accumulation processeses across Greenland with potential applications investigating detailed process models, climate projections, and paleo-climate studies. In the future warmer wetter Arctic (Boisvert and Stroeve, 2015; Schuenemann and Cassano, 2010), changes in the relationship between snowfall and GrIS mass balance may significantly alter the relationships analyzed here. Modeling the evolution of snowfall mass input forward in time would be one way to contextualize these new results more broadly in the uncertain future of GrIS mass balance.

---

*Data availability.*  All data products used in this study are available on the web from the primary sources cited in the acknowledgements.

*Author contributions.*  M.R.G designed the methodology, performed the analyses, and wrote the manuscript. M.D.S. contributed to the methodology development, scientific feedback, and the writing and editing of the manuscript. H.C. contributed remote sensing expertise, scientific feedback, and participated in the production of the manuscript. T.L. contributed feedback on CloudSat precipitation interpretation, helped with the processing of these data, and provided scientific feedback. All authors commented on and approved of this manuscript.

*Competing interests.*  The author(s) declare no competing interests related to the research and writing in this manuscript.

*Acknowledgements.*  This research was supported by the National Science Foundation grants PLR-1303879, OPP-1801477, and PLR-1314156, and the NOAA Physical Sciences Labratory. Thanks to the Laboratoire de Météorologie Dynamique for their support and funding of international collaboration for this work. Thanks kindly to Jen Kay for her support and feedback on topics in this paper. GRACE Mass Variability Time Series' were obtained from NASA's GSFC GRACE Mascon solutions at https://neptune.gsfc.nasa.gov/gngphys/index.php?section=470. CloudSat observations were obtained from the CloudSat Data Center at http://www.cloudsat.cira.colostate.edu/data-products/. NCEP/NCAR reanalysis data were obtained from the NOAA Physical Science Laboratory in Boulder, Colorado, USA (https://psl.noaa.gov/). The authors would also like to acknowledge the many incredible free and open source computing tools that have made this work possible, including but not limited to: Arch (GNU/)Linux, Emacs, NCL, python and the following packages: pandas, scipy, numpy, matplotlib, and scikit-learn. The work in this manuscript is dedicated to the memory of Mr. Wilson, you didn't just teach high school science you taught lessons in how to be a curious person, your caring will not be forgotten.

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
