# Peer review of "Relating snowfall observations to Greenland ice sheet mass changes: an atmospheric circulation perspective"

_The Cryosphere, 2021_

## Author Response (AR1)

Kind thanks to the reviewers and editors for their time and effort in examining our work, the critiques and comments provided were detailed, helpful, and have resulted in a significantly improved manuscript. Almost all changes suggested by reviewer comments were integrated into the revised submission, figures have been modified, and the authors have provided responses directly to comments from individual reviewers where necessary. Thank you also for the additional time it will take you to read these responses, any further commentary is of course welcome and we are happy to clarify as necessary.

**Response to reviewer 1:**

Reviewer one: thank you for your supportive comments on the methodology and results of the study. We agree with the emphasis on reproducibility and accountability and we have revised the manuscript to reflect this. Clear and open presentation of methodology is in the best interest of research and progress, and we've made a clear statement that the code is available to anyone who should be interested. Thank you also for providing detailed comments and technical corrections.

All of the specific comments from reviewer two were integrated into the revised texts, as these critiques were insightful and have improved the text. Below are a few specific comments where important or requested:

Reviewer one: "can you comment on why Fig.3 shows that patterns [b,1] [c,1] and [d,1] occur most often in 'melt' months but that the regression in Fig.6 and Fig.7 seem to have more 'non-melt' months data samples? Is the (top) histogram overlapping melt and non-melt bars or stacking them?"

The histogram is stacking the melt and non-melt bars. The reasons these southerly patterns occur more often in non-melt months is simply because of the larger portion of the year, 5 vs 7 months. While this is a small difference, it's enough to approximately even out the number of both blue and orange points. Although the long-tail in the histogram is entirely orange.

Reviewer one: "Figure 1: I suggest to add the 2km elevation line in Fig.1c"

We had included this in a prior revision, but it was rejected during internal review for being too 'busy'. If the editor would like, we can add this to the figure.

Reviewer one: "It is mentioned (L. 363) that not all months are included in the regression of Fig.6. Could you specify which months that are and why not all months from the GRACE observations are utilized?"

This was poor wording on the part of the author, the idea to be communicated was that the trends quantify the linear relationship between a specific circulation pattern(s) and the mass change that month. While there are clear relationships, the mass change in any given month is also impacted by the circulation patterns that occured in that same month. Meaning a month with 7 [c,1] occurrences but 20 [a,4] occurrences would have a very different relationship to mass balance than a month with 7 [c,1] occurrences but 20 [e,1] occurrences. All available data was included in the figures.

Reviewer one: "Section 4.2.1: I think I am misinterpreting something in the discussion about dynamic mass loss. L. 328 states a bound on dynamic mass loss of 10 to 30 Gt/year, but as this is read from Fig.5 this should be 30 Gt/month? But if that is the case, the comparison to the estimated dynamic loss of 50 Gt/year from literature (L. 336) is no longer 'realistic to the first order'. Could you please check and/or clarify the units presented in this part?"

This was confusing wording on the part of the author, the 21 Gt/month is the correct number from the mass loss in the non-existant year with zero snowfall. However, the average snowfall above 2km is approximately 17 Gt/month. Thus the net mass loss due to dynamics above 2km is  4 Gt/month according to our methodology. In a year, this would be 48 Gt, the wording in the text has been changed to be this clear.

**Response to reviewer 2:**

Reviewer two, thank you kindly for the detailed commentary on both the scientific content as well as the presentation. Thank you also for your clear investigation and supportive commentary on the novel

methodology developed for this analysis.

In the summary comments from reviewer two, the sole concern expressed was for the role of intra-pattern variability of snowfall and its relationship to the quantified snowfall magnitudes. The authors have spent significant time investigating sources intra-pattern variability prior to the submission of this paper and will address these concerns directly here. Before discussing, it is important to clarify that there data is aggregated into two categories for daily circulation patterns, cumulative (net) values and seasonally anomalous values. Sources of intra-pattern pattern variability are significantly different for each. For the cumulative values (Fig. 4) used to quantify net daily mass input, intra-pattern variability is significant and can be almost entirely attributed to the seasonal cycle of variability in snowfall across Greenland. In this way, the snowfall magnitude for each pattern is a spatial and seasonal average of the snowfall mass input. Whereas for anomalies (Fig. 2), having subtracted the annual snowfall cycle from the observations, the primary source of intra-pattern variability is simply only the spatial variance in pixel aggregation (Fig. 1). Thus, there is significantly more intra-pattern variability present in the net values than the anomalies.

Interestingly, to add to the scientific discussion, the authors have previously calculated the net snowfall magnitudes for each season for the circulation patterns and there are many fascinating results. For example: patterns with northerly and weak zonal advection have significantly lower intra-pattern variability than the 'active' southerly regime identified in the paper. Meaning occurrences of these northerly patterns in winter and summer result in approximately the same net magnitude snowfall. Whereas southerly patterns are shown to vary significantly between seasons ( +/- 1Gt) with the most snowfall actually occuring in winter(!), although southerly advection in winter is exceedingly rare. These figures are posted along with this comment.

However, the primary issue in looking at the data this way is that there are significantly lower statistics for each season for some individual nodes due to the intra-annual variability in circulation node frequency. While some qualitative insight can be gained this way, the authors chose not to include these figures in the paper because there are severe questions about the sample size for slicing specific circulation patterns during specific seasons and thus the representativeness/statistical validity of these numbers values is in question. It is for all of these reasons that an analysis of intra-pattern variability, and the subtlety therein, would be a scientific analysis in its own right. In particular, this analysis would be framed as an exploration of the inter-annual variability of snowfall first and foremeost as opposed to quantifying the contribution of snowfall to GrIS mass balance (as was the goal of this paper).

All of the specific comments from reviewer two were integrated into the revised submitted text. We would like to thank them again for their effort in providing this valuable feedback. The authors contribute replies where further warranted.

Reviewer two: "Figure 5: I don't understand why the dots are colored in orange for "melt months". Earlier you stated that little melt occurs above 2 km elevation (L257: "...surface melt generally does not occur."). Even if there is some melt at these high elevations, it's unlikely that it happens in May and September. So the distinction between orange and blue does not make a lot of sense."

Melt months are highlighted in orange so as to complement the discussion in the text about the differences in mass input between summer and winter months. The fact that all large snowfall events and thus mass increases occur in summer (and that the number is quantified directly) is a new and important result. While it may not be relevant to this figure specifically, it's important contextually.

Reviewer two: "L26: Please consider capitalizing "ice sheet". It's the Amazon River, the Tibetan Plateau and should be the Greenland Ice Sheet."

While this is funny, true, and agreeable, we will refer this decision to the copy editors at The Cryosphere.

Reviewer two, asked about the trends and differences in frequency of occurrence for the southerly regime:

While the authors agree that understanding the long-term connections between trends in snowfall and circulation is a fascinating question, it's unfortunately untenable with the available data. The authors have tried using several methodologies to identify trends in the occurrence of important regional circulation patterns and have failed to reach convincing conclusions. While the 70 year reanalysis record of surface

pressure provides enough data to identify changes in circulation beyond decadal cycles of the AO/NAO, connecting those trends to the ¡20 year record of snowfall is a difficult proposition. Unfortunately this is still the realm of modelers and is impossible to tie to available CloudSat observations. Further, the 'large' suite of circulation patterns (5x4 SOM) is compelling for its ability to provide a more detailed look into circulation variability. Even with the long reanalysis record, the authors have found that identifying trends still requires a more blunt tool and is statistically limited to a more granular search space of 6 (3x2) or less patterns. This means that any of the interesting spatial variability identified in this paper would be significantly less clear.